# Exploiting Biological Nitrogen Fixation: A Route Towards a Sustainable Agriculture

**DOI:** 10.3390/plants9081011

**Published:** 2020-08-11

**Authors:** Abdoulaye Soumare, Abdala G. Diedhiou, Moses Thuita, Mohamed Hafidi, Yedir Ouhdouch, Subramaniam Gopalakrishnan, Lamfeddal Kouisni

**Affiliations:** 1AgroBioSciences Program, Mohammed VI Polytechnic University (UM6P), Benguerir 43150, Morocco; hafidi.ucam@gmail.com (M.H.); youhdouch@gmail.com (Y.O.); Lamfeddal.KOUISNI@um6p.ma (L.K.); 2Laboratoire Commun de Microbiologie (LCM) IRD/ISRA/UCAD, Centre de Recherche de Bel Air, Dakar 1386, Senegal; 3Département de Biologie Végétale, Faculté des Sciences et Techniques, Université Cheikh Anta Diop (UCAD) de Dakar, Dakar 1386, Senegal; 4Centre d’Excellence Africain en Agriculture pour la Sécurité Alimentaire et Nutritionnelle (CEA-AGRISAN), UCAD, Dakar 18524, Senegal; 5International Institute of Tropical Agriculture, Nairobi PO BOX 30772-00100, Kenya; M.Thuita@cgiar.org; 6Laboratory of Microbial Biotechnologies, Agrosciences and Environment, Faculty of Sciences Semlalia, Cadi Ayyad University, Marrakesh 40000, Morocco; 7International Crops Research Institute for the Semi-Arid Tropics (ICRISAT), Patancheru 502319, India; s.gopalakrishnan@cgiar.org

**Keywords:** BNF, biofertilizers, legumes, yield improvement, inoculum quality, biofertilizer market

## Abstract

For all living organisms, nitrogen is an essential element, while being the most limiting in ecosystems and for crop production. Despite the significant contribution of synthetic fertilizers, nitrogen requirements for food production increase from year to year, while the overuse of agrochemicals compromise soil health and agricultural sustainability. One alternative to overcome this problem is biological nitrogen fixation (BNF). Indeed, more than 60% of the fixed *N* on Earth results from BNF. Therefore, optimizing BNF in agriculture is more and more urgent to help meet the demand of the food production needs for the growing world population. This optimization will require a good knowledge of the diversity of nitrogen-fixing microorganisms, the mechanisms of fixation, and the selection and formulation of efficient *N*-fixing microorganisms as biofertilizers. Good understanding of BNF process may allow the transfer of this ability to other non-fixing microorganisms or to non-leguminous plants with high added value. This minireview covers a brief history on BNF, cycle and mechanisms of nitrogen fixation, biofertilizers market value, and use of biofertilizers in agriculture. The minireview focuses particularly on some of the most effective microbial products marketed to date, their efficiency, and success-limiting in agriculture. It also highlights opportunities and difficulties of transferring nitrogen fixation capacity in cereals.

## 1. Introduction

From 1.6 billion in 1900, the world’s population has grown to more than 7 billion today and will reach 9 billion by 2050 [1,2]. In this context, it will be impossible to feed the world’s growing population without significant increase in the agricultural production. Some authors argue that one of the best ways to accelerate world agricultural production is to apply inorganic fertilizers specially nitrogen [3,4,5,6,7]. Indeed, crop production is dependent on nitrogen (*N*) which is a limiting factor, and the gap between its supply and demand is continuously growing [1,8,9]. On the other hand, excessive use of inorganic nitrogen fertilizers has led to ecosystem perturbations across the world [10,11,12]. This justifies the emerging demand to reduce the systematic use of inorganic *N* fertilizers and promote sustainable agricultural and agroforestry practices [13,14]. Among alternative approaches, exploiting biological nitrogen fixation (BNF) appears as a route to reduce the input of *N* fertilizers in agriculture and thereby their negative environmental impacts. In fact, BNF is a natural process of changing atmospheric nitrogen (N_2_) into a simple soluble nontoxic form (NH_4_+ primarily) which is used by plant cell for synthesis of various biomolecules. Nitrogen fixation is one of the major sources of nitrogen for plants and a key step distributing this nutrient in the ecosystem [15,16]. Biological nitrogen fixation is performed exclusively by prokaryotes: archaea and bacteria. For bacteria, different groups are involved, including free-living bacteria belonging to genera such as Azotobacter, Azospirillum, Bacillus, or Clostridium; symbiotic bacteria like Rhizobium associated with legumes; Frankia associated with actinorhizal plants; and cyanobacteria associated with cycads [17,18]. For archaea, nitrogen fixation is still restricted to groups that produce methane, called methanogens [19].

Nitrogen-fixing organisms can be classified into three categories: free-living *N* fixers, associative *N* fixers, and symbiotic *N* fixers. The last two groups can be found in the rhizosphere of legume and non-legume plants [20,21]. Nevertheless, root nodule symbiosis is one of the most studied mutualistic relationships of plants and nitrogen-fixing organisms. Root nodule symbiosis is also the most effective in *N*-fixing (20–300 Kg ha^−1^. an^−1^) and the more important because it involves almost all food and fodder legumes. The establishment of this mutualistic relationship starts with a molecular dialog between the two partners, host plant and nitrogen-fixing organism through the flavonoids and isoflavonoids secreted by the host plant in its rhizosphere [22,23]. The molecular dialog allows recognition, infection, differentiation of root hair cells, and nodule development. Inside nodules, symbiotic bacteria, in a form called bacteroides, fix nitrogen [24,25]. Among the root nodule symbioses, the legume-rhizobium association model has received most attention because several legumes are food or cash crops, and some have been used to select effective strains of nitrogen-fixing bacteria for biofertilizer production. In this respect, significant advances are being made to develop inoculums containing effective nitrogen-fixing bacteria, particularly for poor soils with zero or low input of *N* fertilizers [26,27,28,29]. Moreover, in the last few years, significant efforts have been made to extend nitrogen fixation to crops other than legumes, particularly cereals [30,31]. Compared to symbiotic nitrogen-fixing bacteria, non-symbiotic bacterial diazotrophs have limited agronomic significance, although their contribution is estimated at about 30% of total BNF [32,33] and can be a significant fixed *N* source in many terrestrial ecosystems [31,34]. This potential has been proven by the results of Pankievicz et al. [35] who showed *Setaria viridis*, inoculated with an ammonia excreting strain of *Azospirillum brasilense* showed robust growth under nitrogen-limiting conditions. Recent work from Van Deynze et al. [36] have shown that a Mexican maize landrace can fix nitrogen at a rate of up to 82% when it is associated with non-symbiotic diazotrophs bacteria present in its mucilage of aerial roots.

The objective of this minireview is to highlight the agronomic importance of BNF, a non-polluting and cost-effective way to improve soil fertility and crop production. It presents summary of major processes of nitrogen cycle, nitrogen fixation mechanisms, and the contribution of BNF to agriculture. It then focuses on commercial products with *N*-fixing bacteria as biofertilizer and concludes with research perspectives.

## 2. Plant Available Nitrogen

In soil, nitrogen is found always in two major forms: inorganic, as mineral nitrogen (~2%), and organic (~98%). Inorganic forms include ammonia (NH_3_), ammonium (NH_4_^+^), nitrite (NO_2_^−^), and nitrate (NO_3_^−^), while organic forms are found in living organic matter (soil biota and fresh animal and plant debris) and non-living organic matter including humified and non-humified compounds. Mineral nitrogen is available to plants in two forms, either as ammonium nitrogen (NH4+-N) or as nitrate-nitrogen (NO3-N). Organic nitrogen is not directly available to plants and must be converted through a slow process (mineralization) to ammonium or nitrate [37]. Once available, nitrogen is subject to strong competition between plants and microorganisms. In addition, *N* is continually lost through soil erosion, denitrification, leaching, chemical volatilization, and perhaps most importantly, removal of *N*-containing crop residues from the land [2,38]. In this respect, nitrogen is often in short supply in many croplands, limiting crop growth and productivity. Synthetic nitrogen fertilizers have been introduced to compensate *N* deficiency in agricultural soil. The Haber–Bosch process, developed in 1913, is still the main industrial procedure for the production of ammonia (NH_3_), by combining atmospheric nitrogen (N_2_) with hydrogen (H_2_). The produced ammonia (NH_3_) can then be used to make numerous other nitrogen compounds such as nitrate, ammonia, ammonium, and urea [2]. This process adds more reactive nitrogen to the global nitrogen cycle consisting of nitrogen fixation, assimilation, mineralization, nitrification, and denitrification [16,39,40]. Furthermore, the production of nitrogenous fertilizers requires huge quantities of fossil fuels, which represent ~2% of the world’s energy consumption [41]. Unfortunately, substantial amount of nitrogen applied to the soil is not absorbed by crops. Almost 25% of the nitrogen supplied as fertilizer is lost through leaching and other factors during various agricultural processes [15]. These cumulative effects are evident as higher waste and pollution which adversely affecting soil health and environment in general [42].

Biological nitrogen fixation does not require fossil fuels, and thus is an environmentally friendly source of *N* for crop production. The fixed nitrogen is less susceptible to denitrification, leaching, and volatilization because it is directly absorbed by plants [43,44]. Therefore, optimization of BNF is critical to sustain both food production and environmental health. To achieve this goal, it is essential to identify elite strains of nitrogen-fixing organisms and include more legumes in agroecosystems for efficient BNF [3,45].

## 3. Nitrogen Fixation

Biological nitrogen fixation converts di-nitrogen (N_2_) into plant-usable form (NH_4_+ primarily). The process consists of combining N_2_ with the hydrogen ions from water. N_2_-fixation is not only a biologically-mediated process because lightning or fire can also oxidize N_2_ to nitrate (NO_3_^−^). Lightning makes ~1% ammonia of the net nitrogen fixed per year [46]. All organisms (eukaryotes and prokaryote) naturally depend directly or indirectly on BNF for their *N* supply. This *N* is the main element for the synthesis of nucleic acids, proteins, and other organic nitrogenous compounds. Biological nitrogen fixation is an energetically expensive process because 16 ATP molecules are needed to break down an N_2_ molecule. Twelve additional ATP molecules are required for NH_4_^+^ assimilation and transport, totaling 28 ATP molecules. The nodulating plants must provide 12 g of glucose to their bacterial partners to benefit 1 g *N* in part [47]. However, this process is still less energetically expensive than the Haber–Bosch process, developed in 1913. To produce the same amount of nitrogen, the Haber–Bosch process requires a temperature of 400–500 °C and a pressure of ~200–250 bars [48].

N_2_ fixation is catalyzed by nitrogenase, which is quite similar in most of the nitrogen-fixing bacteria. Nitrogenase is an enzyme complex with two metal components: dinitrogenase MoFe (molybdenum-iron protein) serving as the catalytic component and dinitrogenase reductase (Fe protein). These two metal components are encoded by the *nif* genes, the *nif*D and *nif* K genes coding for MoFe dinitrogenase and the *nif* H gene coding for Fe dinitrogenase reductase [49,50,51,52]. In addition to nitrogenase, several regulatory proteins involved in nitrogen fixation are encoded by *nif* genes. Depending on the requirements of molybdenum (Mo), vanadium (V), or iron (Fe), there are three different forms of nitrogenase. Each nitrogenase contains an active site for the reduction of the substrate and this site is composed of a complex metal group called FeV-cofactor, FeFe-cofactor, and FeMo-cofactor, for, respectively, V-nitrogenase, Fe-nitrogenase, and Mo-nitrogenase [53]. Whatever the type, nitrogenase is inactivated in aerobic environment because it is extremely O_2_ sensitive. Indeed, oxygen inactivates and destroys nitrogenase and has an inhibitory effect on nitrogen fixation and assimilation pathways [54]. The Fe-nitrogenase and V-nitrogenase are particularly sensitive to oxygen, while Mo-nitrogenase is slightly less susceptible [55].

Some nitrogen-fixing microorganisms have evolved various strategies to avoid the inhibitory or toxic effect of oxygen. For instance, many diazotrophic bacteria fix N_2_ only under anaerobic or microaerophilic conditions. In aerobic chemotrophs and phototrophs which need access to oxygen or produce it as part of their metabolism, these bacteria manage to achieve a good trade-off between the efficiency of using O_2_ as an acceptor of electrons and the inactivation of nitrogenase [56]. The mechanisms in cyanobacteria (free-living photosynthetic) is to separate the O_2_ they produce from their nitrogenase system [57]. Thus, some groups of cyanobacteria belonging to Nostoc and Anabaena genera develop heterocyst as specialized cells for nitrogen fixation. The heterocyst has thick cell walls which protect the dinitrogenase enzyme complex against O_2_. In other non-heterocyst cyanobacteria, there is a temporal separation between the N_2_ fixation and O_2_ production. The fixation of nitrogen is achieved during darkness in the absence of O_2_ production [58].

To maintain a low oxygen concentration inside the cell, some bacteria such as Azotobacter express a high respiratory rate [59]. At high O_2_ level, they can change the conformation of the nitrogenase in protected inactivated state and prevents its irreversible damage [60]. Sabra et al. [61] showed alginate capsules formed on the surface of cells play an important role for survival of diazotrophically growing *Azotobacter vinelandii* under aerobic conditions. They proposed alginate production as a new mechanism for protecting nitrogenase against oxygen. Indeed, alginate polymers act as a barrier against oxygen and reduce its transfer into the cell.

## 4. Retrospect on the Isolation of Nitrogen-Fixing Bacteria and Launch of *N*-Fixing Biofertilizers

### 4.1. Discovery

Biological nitrogen fixation was discovered by Hellriegel and Wilfarth (1886), who reported that some legumes could use nitrogen gas (N_2_) from nodules on their roots [62]. Two years later (1888), the *N*-fixing bacteria strain *Rhizobium leguminosarum* was isolated for the first time by Beijerinck [63]. In 1893, Winogradsky reported the isolation of *Clostridium pasteurianum* [64], while the concept of inoculation of legumes with *N*-fixing rhizobia was introduced to New South Wales (Australia) farmers by Guthrie in 1896 [65]. Five years later (1901), the aerobic heterotroph Azotobacter has been isolated and described by Beijerinck. The first field trials using a culture of rhizobium and field peas have been carried out in 1905 in Australia (Hawkesbury agricultural college) and later in 1914, farmers have been supplied with rhizobial inoculants [66]. In 1953, Johanna Döbereiner’s work has afforded new insights into the BNF with the identification of a new category of nitrogen fixing endophytes: *Beijerinckia fluminensis* associated with sugarcane and *Azotobacter paspali* associated with Bahia grass (*Paspalum notatum*) [67,68]. At the end of 1975, a nitrogen-free medium called NFb (Fb stands for Fabio Pedrosa) was developed by Fabio Pedrosa for the isolation of Spirillum species. Therefore, two species of Azospirillum (*A. lipoferum and A. brasilense*) have been isolated by using the semi-solid NFb medium [68,69]. These two species are heterotrophic and non-symbiotic bacteria that can fix 20–30 kg *N*/ha/year on average [70]. In 1979, Steward used the Nitrogen-15 (^15^N) tracing technique to demonstrate nitrogen fixation in cyanobacteria (formerly blue-green algae) [71]. For archaea, their ability to fix nitrogen was only recently highlighted by independent discoveries of diazotrophic growth in two different methanogenic archaea: *Methanosarcina barkeri* [72,73] and *Methanococcus thermolithotrophicus* [73,74]. The ^15^N tracing technique and acetylene reduction assay (ARA) technique have confirmed nitrogen fixation in *M. barkeri* and *M. thermolithotrophicus*, respectively. Some major events in the history of research on nitrogen and BNF are summarized in Table 1. Currently, at least thirteen genera belonging to prokaryotes group are known to fix nitrogen (Figure 1).

This ability to fix nitrogen is increasingly reported in groups that were not expected to. Recently, by analyzing 16,989 *nif* H sequences of nitrogen-fixing microorganisms, Gaby and Buckley [80] have concluded that diazotrophic diversity is poorly described and many organisms still to be discovered. The discovery of many nitrogen-fixing microorganisms has subsequently led in few years to the development of numerous commercial biofertilizers across the world. Nitragin, made with *Rhizobium* strains, was the first commercial biofertilizer launched in the USA by Nobbe and Hilther in 1895 [81]. During the 1900s Nitragin was followed in the market by Azotogen (*Azotobacter chroococcum*) and Phosphobactin (*Bacillus megaterium* cv phosphaticum) launched in Russia. In India, commercial production of nitrogen biofertilizer was initiated by the ICAR-Indian Agricultural Research Institute (IARI), and Agricultural College and Research Institute in 1956 [82]. In Africa, since 1977, the Microbial Resource Center (MIRCEN) has contributed to the transfer of BNF knowledge to researchers, extension workers, and farmers. In this respect, in the late 1990s many public and private organizations in eastern and southern Africa have been involved in the production of *N*-fixing inoculants [83].

### 4.2. Commercialization

Currently, many biofertilizer products exist over the world; however, the nitrogen-fixing biofertilizer market controls the largest part in global biofertilizer market. From USD 800 million in 2016, it is anticipated to be USD 3 billion by the end of 2024 [84]. In the global biofertilizer market, North America (USA, Canada, and Mexico) holds the largest share accounting for around 27.7% (Figure 2A). For instance, there are more than 150 microbe-based biofertilizers in Canada, most of which are based on nitrogen-fixing bacteria, i.e., Rhizobium strains for legumes. Europe (Germany, UK, Spain, Italy, and France) occupies the second place in terms of biofertilizer production, with around USD 0.45 billion dollars by 2017. The third largest biofertilizer market is Asia-Pacific (China, Japan, India, Australia, New Zealand, and rest of Asia) with USD 0.284 billion in 2017 and USD 0.44 billion by 2018 [82]. China holds more than 511 biofertilizers products and accounted for 43.2% of the biofertilizers market share for Asia-Pacific region in 2017 [85]. South America biofertilizers market was valued at USD 0.239 billion in 2017 and Brazil holds the largest share, with USD 0.135 billion. For Africa, the biofertilizers market is still small (USD 0.0445 million in 2017). The main biofertilizer producing countries are South Africa with USD 0.0293 billion of market value, Egypt and East African countries such as Uganda, Kenya, Tanzania, and Sudan. According to the Global Biofertilizers Market [84], Europe and Latin America are currently the top consumers of biofertilizers followed by China and India, because in many countries from these regions, there are stringent regulations imposed on chemical fertilizers. In the worldwide market for biofertilizers, the phosphate solubilizing biofertilizers (with a share of 14%) occupy the second place far behind the nitrogen-fixing biofertilizers (accounting ~79%), and the other types of biofertilizers hold the remaining 7% (Figure 2B).

## 5. BNF and their Contribution to Agriculture

### 5.1. Inoculants for Legume Crops

*N* input through BNF is approximately 122 million tons of *N* per year of which 55 to 60 million tons is fixed by agricultural crops [86,87,88]. Soybean (*Glycine max*) legume has the highest contribution of BNF; this species fixes annually ~16.4 million tons of *N* [89]. The main microsymbionts of soybean belong to *Bradyrhizobium* species [90].

The potential of BNF providing nitrogen *N* in ecosystems is increasingly being exploited in agricultural practices, mostly through legume cultivation (Soybean, lupin, alfalfa, chickpea, cowpea, etc.). Legume–rhizobium symbiosis is an important facet of symbiotic nitrogen fixation [91,92]. Inoculation of legumes crops with Rhizobia is one of the success stories of biofertilizers in agriculture. The positive impact of diazotrophic microorganisms on agriculture has opened the biofertilizer market. In few years the biofertilizer market has grown and at present, many nitrogen-fixing microorganisms are marketed as biofertilizers (Table 2). Different products are available and some of them have shown great potential by improving crop growth and yield and could significantly reduce a farmer’s fertilizer bill (Table 2). For example, in Brazil, the economic benefit in terms of *N*-fertilizer saving was over USDA 2.5 billion per year by 2002 [93]. Use of BNF-based commercial inoculums has contributed to increase soybean yield in Brazil, and therefore helped to put the country in second place among the largest soybean producers behind the USA. In the USA, the contribution of BNF to the soybean *N* nutrition ranged from 23 to 65% [94]. In Spain, Pastor-Bueis et al. [95] showed that *Rhizobium leguminosarum* bv. *phaseoli* LCS0306, formulated with perlite-biochar carriers, produced a significantly higher grain yield of common bean (3640 kg ha^–1^ versus 3165 kg ha^–1^ in the *N*-fertilized control plot). In Ghana, Ulzen et al. [96], by comparing urea application to two commercial biofertilizers (Biofix and Legumefix) on soybean and cowpea, reported that these inoculants were more profitable. They increase nodule dry weight (>two-fold), nodule number (90–118%), and grain yield (12–19%) compare to the control (urea). In northern Nigeria, Ronner et al. [97] also showed that soybean inoculation with rhizobia has increased yield by 447 kg/ha compared to the control. Similar results were reported by Thuita et al. [98] who recommended for sustainable soybean yield increase, to inoculate with Legumefix + sympal (a fertilizer blend for use with rhizobia inoculants) or biofix + sympal to raise yields from 2000 kg/ha to 4000 kg/ha. In poor soils, amendment with vermicompost in addition to Sympal and Legumefix has been shown to improve soybean yields [99]. Previously, in a study of several commercial rhizobial inoculum, Thuita et al. [100] reported that these products have potential to increase growth, yield, and nitrogen fixation legumes. A noteworthy contribution of the use of legume inoculants was also reported in the Zambian’s economy, with an input of more than US $23 million in eight years [101]. Recently “Nitragin” a pure culture of root-associated bacteria was improved and tested on soybeans and soyfoods in Germany. Results showed that soil inoculated with Nitragin gave a 3- to 4-fold increase in yield, plus an increase in protein in the roots and leaves [102].

### 5.2. Inoculants for Non-Legume Crops

Several non-leguminous plants, mainly cereals, have developed multiple strategies in association with diazotrophs to cope with *N* deficiency. Some of these microorganisms have been used to make bacterial inoculants. Mexico was one of the first countries to commercialize maize seeds coated with *Azospirillum* [103], followed by Argentina. Field experiments in Sierra Mixe (region of Oaxaca, Mexico) using ^15^N natural abundance or ^15^N-enrichment assessments over 5 years indicated that atmospheric nitrogen fixation contributed to 29–82% of the nitrogen nutrition of maize [36]. In Egypt, El-Sayed et al. [104] showed significant increases (24.8 and 27.2% in the first season and 18.4 and 22.0% in the second season respectively compared to the un-inoculation) in grain yield of barley after inoculation with biofertilizers (Microbin and Azottein, constituted of a mixture of *P*-dissolving and *N*-fixing bacteria), and these results were comparable to those obtained with chemical fertilizers. More recently, Rose et al. [105] demonstrated that a commercial biofertilizer product known as “BioGro” (Table 2) can replace 23 to 52% of *N* chemical fertilizers without loss of yield in rice systems, in Southeast Asia. BNF contribute ~30 kg *N* ha^−1^ per year to rice systems [106]. According to Serna-Cock et al. [107], applying *Azospirillum brasilense, Azotobacter chroccocum,* and *Trichoderma lignorum* as biofertilizer in sugarcane plants (variety CC 934418) can replaces 60% of the nitrogen needed by this cultivar. Corroborating these results, Antunes et al. [108] showed that inoculation with *H*erbaspirillum seropedicae**, *Pseudomonas* sp., and *Bacillus megaterium* increase sugarcane (variety RB92579) yield from 18% to 57.31%. Although cereals benefit significantly from diazotrophs, most microbes are unlikely to fix nitrogen in the presence of high rate of chemical fertilizers.

### 5.3. Success-Limiting Factors of BNF Application in Agriculture

The BNFs have the capacity to reduce the use of nitrogen fertilizers to ~0.160 billion tons per year, which corresponds to a reduction of 0.270 billion tons of coal consumed in the production process [109]. All these results show that BNF is directly proportional to agricultural sustainability. Despite the advantages of microbial inoculant technology, there still exist some success-limiting factors against a universal utilization. In fact, the efficiency of microbe-based biofertilizers depends on many factors including the targeted crop, edaphic (pH, salinity, and soil type), biotic (competition between introduced and indigenous strains, microbial parasites and predators), and climatic factors [110,111] that can make commercial inoculum counter-productive. Besides competition among microbial strains for resources and plant nodulation, partner fidelity and specificity mediated by genetic and molecular mechanisms are among the success-limiting factors against a universal utilization of microbial inoculants [112,113]. On the other hand, commercial inoculants were often made with one or at most two trains, while under field conditions, plants are associated with many strains which provide them diverse benefits through functional complementarity. Nevertheless, the poor performance of biofertilizers is primarily linked to inappropriate strains and inefficient production technology. Herrmann et al. [114], studying the microbial quality of 65 commercial inoculants manufactured in seven different countries, showed that only 36% of the products could be considered as “pure”. Among the remaining 64% some contained one or several strains of contaminants and some products did not contain any strains. However, the study does not specify the origin of this problem. Is it a loss of viability during the storage time or the quality of the product delivered by the manufacturer? Similarly, In India, the evaluation of the quality of legume inoculants showed that most of the products tested did not contain the optimal amount of rhizobium (< 10^8^ rhizobia/g of inoculant) and were contaminated by a large amount of non-rhizobial organisms [115]. Therefore, it is a big challenge to maintain viability and purity of microbes in microbial inoculant [116]. In many regions across the world, farmers are not yet familiar with this type of fertilizer which is sensitive to temperature, humidity, time, and storage conditions; that is why they are sometimes confused about quality and expiry dates of biofertilizers [117]. In Africa, the International Institute of Tropical Agriculture (IITA) has been working with regulatory authorities for biofertilizers in Kenya, Uganda, Tanzania, Ethiopia, Nigeria, and Ghana to establish standards for both registration and efficacy testing to protect farmers from fraudulent products in the market. Previously, N_2_Africa and MIRCEN worked together in order to test commercial inoculants and offer quality assurance to their distributors and customers. In this respect, there is a need, particularly in Africa, to strengthen farmers’ capacities and establish networks for sharing reference protocols and information about BNF. Furthermore, very few firms in many African countries are involved in inoculum production and commercialization, limiting therefore access at adequate timely to quality inoculants.

### 5.4. Beneficial Mechanisms Other Than N-Fixation Provides by Diazotrophs Bacteria

In addition to their *N*-fixing abilities, diazotrophic bacteria are now recognized as also promoting plant growth (PGP) and yield and causing positive changes in soil structure and microbial community [121,122,123]. Many diazotrophic stains belonging to Rhizobia, Bradyrhizobia, Ensifer, Azotobacter, Azospirillum, Pseudomonas, Klebsiella, and Bacillus genera were reported to enhance the plant growth and grain yield of chickpea, bean, pea, wheat and rice through phytohormones and secondary metabolites production [123]. For instance, recent results from Gopalakrishnan et al. [124] have shown that rhizobia act also as PGP by producing indole acetic acid (IAA), siderophores, and organic acids, which leads to a stimulation of stems and roots growth of chickpea (*Cicer arietinum* L.). Some Bradyrhizobial strains isolated from rice rhizosphere and *Azorhizobium caulinodans* associated with *Sesbania rostrate* are capable of fixing nitrogen in the free-living state [125] under low-oxygen conditions [126]. Mia and Shamsuddin [127] have reported beneficial effects of rhizobium inoculation on different cereal crops as rice, maize, and wheat. On the other hand, Gopalakrishnan et al. [128] and Das et al. [129] reported that rhizobia can also act as biocontrol agents against pathogenic fungi (Rhizoctonia, Fusarium, Macrophomina, and Sclerotium), through hydrocyanic acid (HCN), antibiotics and/or mycolytic enzymes. The PGP traits of numerous other *α*-, *β*-, and *γ*-Proteobacteria inhabitants of legume nodules and contributing to N_2_ fixation were always neglected. These new aspects of diastrophic bacteria, especially rhizobia are avenues for research in order to select efficient BNFs, for their better contribution in crop yield.

### 5.5. Synergistic Benefits

Soil microorganisms such as arbuscular mycorrhizal fungi (AMF) are known to have significant positive effect on BNF by direct and/or indirect interaction with *N*-fixing microorganisms. Indeed, AMF play a significant role in uptake of water and nutrients from soil [130] necessary to generate energy required for BNF [131] Moreover, through their hyphal networks, AMF can facilitate the colonization of legume roots by symbiotic *N*-fixing bacteria [132], as well as the transfer of nutrients and symbiotically fixed *N* between similar or dissimilar plants [133,134]. On the other hand, bacteria can also be beneficial to AMF. After characterizing a commercial AMF inoculum (AEGIS, i.e., Atens, Agrotecnologias Naturales S.L), Agnolucci et al. [135] showed that this product harbors many bacteria with important functional PGP properties such as nitrogen fixation, inorganic phosphate solubilization AIA production, etc. The synergic effects between AM fungi and soil microbial communities increase plant biomass and N acquisition from organic matter.

## 6. Challenges of Extending the BNF Ability to Non-Legumes

Cereal production is underpinned by high nitrogen fertilization from chemical fertilizers. However, the excessive use of this nitrogen source leads to soil acidification [136]. Therefore, for several years scientists have tried to find a way to reduce dependence on chemical fertilizers through engineering crop plants to fix nitrogen they need for their growth and yield [49,134,136,137]. This process was restricted primarily to legumes in agricultural system. The idea to transfer nitrogen fixing capacity to non-fixing crops, such as wheat, rice, sorghum, or maize, is one of the long-standing dreams of many researchers in plant biology. Many studies have been done during the past years in an effort to determine the kinds and species of soil microorganisms that possess the ability to fix N_2_-nitrogen, to characterize factors encouraging plant colonization by *N*-fixing bacteria [138,139], and to identify bacterial genes that encode nitrogenase [140]. However, transferring BNF traits to non-legumes, especially cereals, remains elusive. Research focuses on mostly two strategies to implement this objective.

The first is to engineer new symbioses between cereals and *N*-fixing bacteria by transferring the genes of legume plants necessary for the development of root nodule symbiosis to cereals [21,49]. This approach requires a genetic modification of the plant to release nodulation signals. However, the main difficulty is to deal with toxic effect of oxygen, because nitrogenase requires an anaerobic environment within the cell in order to function successfully. In addition, nitrogen fixing symbiosis requires the coordinated function of more than 30 essential genes [141].

The second approach is to directly introduce nitrogenase into the cereal plant cells [142,143]. This approach will allow the plant to synthesize its own nitrogen without the need for bacterial interactions. However, to achieve this goal, it is necessary to synthesize nitrogenase in cereals. The complexity of this biosynthesis and the sensitivity of this enzyme to oxygen is a major challenge for the implementation of this strategy. In addition, it is unclear whether cereals host can provide the reducing power and energy needed to sustain nitrogenase catalysis [144].

Despite scientific and technological progress, the transfer of nitrogenase to cereals remains an object to be achieved because both approaches face technical problems and make difficult to implement the different strategies. However, in view of the genetic advances and the successful transfer of nitrogen fixation (*nif*) genes to *E. coli*, to *Saccharomyces cerevisiae* (eukaryotic model organism), and to plastids of tobacco, there is a hope that these objectives might be realized in the near future [49]. However, more intensive collaborative research and an international coordination are required.

The combination of recent advances in comparative genetic analysis and synthetic biology has allowed tremendous progress toward the objective of engineering nitrogen fixation to non-legumes. Scientific efforts of several years of research around the world have resulted in important results like the sequencing of nitrogenase genes like *nif* H, *nif* D, *nif* K, *nif*E, *nif*N, etc. and the creation of *nif* database in 2012 [80]. Owing to this database, which contains 32 954 sequences to date, we have a better understanding of the evolutionary history of nitrogenase. On the other hand, a comparative analysis of the symbiotic systems of non-legumes, *Parasponi*, legumes, and Actinorrhizae allowed to identify the core genetic networks underlying root nodule formation and functioning [145,146,147] and to define strategies for transferring nitrogen-fixing ability to non-legume crops [20]. On the other hand, to overcome the obstacle of oxygen sensitivity, a promising solution could come from cellular organelles like mitochondria and root plastids. Indeed, these two organelles offer a low-oxygen environment and therefore suitable for nitrogenase expression in eukaryotes [146,148]. Furthermore, both organelles can provide high concentrations of adenosine 5′-triphosphate and reducing power required for nitrogenase activity [49] and they are similar to prokaryotes in terms of gene organization and expression. In this respect, Ivleva et al. [149] have produced the active Fe protein of nitrogenase in transgenic plants after successfully integration of *nif*H and *nif*M genes into the tobacco chloroplast genome. Burén et al. [49] have identified a minimum *nif* cassette of nine genes sufficient for nitrogen fixation that they have already tested in transgenic yeasts (*Saccharomyces cerevisiae*). These results bring us closer to the goal in engineering an active nitrogenase enzyme in plants. Table 3 summarizes some successful steps already achieved.

Besides these approaches, other researchers continue to improve the nitrogen fixation pathway in diazotrophic endophytic, associative, and symbiotic organisms that are already in relationships with plants [156,157,158,159] by using different strategies and tools. Among these strategies, we can list the optimization of the colonization process, the optimization of carbon supply from root cells to endosymbiotic bacteria, engineering of respiratory protection and O_2_-binding proteins to allow aerobic nitrogen fixation by microsymbionts, and improvement of ammonium uptake by plant cells [21]. For instance, in a low pH environment, some strategies are suggesting treating plants with flavonoids, Nod Factors, or phytohormones to overcome negative effects of low pH [160,161,162]. Other strategies try to generate acids tolerant legumes cultivars and rhizobia strains for enhancing symbiotic nitrogen fixation [163]. On the other hand, significant progress has been made to understand the biochemical, physiological, and ecological aspects of diazotroph associations with cereals. For example, diverse communities of endophytic bacteria have been identified in sugarcane (*Saccharum *spp.), Oaxaca maize (*Zea mays*), sweet potato (*Ipomoea batatas* L.), and paddy rice (*Oryza sativa* L.) rhizosphere and their agronomic significance have been proved [164,165]. In addition to the traditional approach of identifying single-strain isolates with a range of advantages for plants, several studies propose to develop synthetic consortia of bacteria with resilient functionality in terms of promoting plant growth [166]. A genetic and genomic analysis of selected strains will allow targeted modification of the bacterial genomes in order to confer better advantages on the crops [167].

## 7. Road Map for Successful and Large-Scale Adoption of *N*-Fixing Biofertilizers

Currently, agricultural production depends on the large-scale use of chemical fertilizers [168]. To take full advantage of BNFs for improving soil health and crop quality, different questions should be addressed by researchers and other stakeholders through an integrated and collaborative framework. Here, we provide a summary of research and development priorities needed for successful and large-scale adoption of *N*-fixing biofertilizers.

Better understand the mechanisms of biological nitrogen fixation, as well as the mechanisms of partner choice and partner fidelity.

Broaden the host spectrum of symbiotic bacteria by reducing the symbiotic specificity through good comprehension of the genetic and molecular mechanisms that regulate symbiotic specificity.

Make rigorous testing of inoculums under a wide range of environmental conditions and soil types before its commercialization [169].

Intensify the efforts towards identifying highly competitive and effective microbial strains, with particular attention to mixed-strain consortiums rather than mono-strain inoculums to take advantage of functional complementarity under field conditions.

Improve the quality of commercial inoculums with emphasis on strain purity, inoculum density, and formulations that better preserve the vitality of microbial strains during storage. The performance of microbial strains should be also considered as a factor of inoculant quality.

Strengthen quality control of commercial inoculums by imposing official registrations and controls by independent third-party services.

Strengthen farmers’ capacities through training and the establishment of networks for sharing reference protocols and information about BNF.

Make easy access to quality inoculants timely for end users and develop new inoculation methods.

Associate mycorrhizae with BNF will facilitate the transfer of nitrogen from plants with high fixing potential to low or non-fixing plants because it is already established that the presence of arbuscular mycorrhizal increase the transfer of symbiotically fixed *N* through network connection between similar or dissimilar plants [133,134].

Identify soils suitable for inoculation with *N*-fixing organisms, with particular attention to low pH and high levels of *N* in soil which are known to inhibit the formation of nodules. By superimposing nitrogen fertility map to pH map [170,171], we identified different areas with high potentials for inoculation success due to their soil pH (ranges from slightly acid to neutral, 5.5 to 7.2) and relatively low levels of nitrogen in soil (Figure 3). For instance, Africa appears as an ideal continent for a large application of *N*-fixing biofertilizers.

## 8. Conclusion and Perspectives

This review highlights the agronomic importance of BNF, a non-polluting and cost-effective way to improve soil fertility and crop production. It should be emphasized, however, that the successful and large-scale adoption of BNF mostly depends on understanding the factors that control the BNF systems in the field conditions, improving the quality of commercial inoculums, and strengthening farmers’ capacities. Engineering the capacity to fix nitrogen in cereals, either by themselves or in symbiosis with nitrogen-fixing microbes, represent attractive future options that, nevertheless, require more intensive and internationally coordinated research efforts. Biotechnology of BNF is indeed an opportunity to help close the yield gap in over the world and especially for underdeveloped country, and thus it appears necessary to integrate BNF in plant breeding programs. Although hurdles are associated with the large-scale commercialization of nitrogen-fixing biofertilizers, this biotechnology remain promising option for healthy and sustainable agriculture with low dependence on industrial nitrogen production. Our analysis provides a road map for successful and large-scale application of *N*-fixing biofertilizers.

## Figures and Tables

**Figure 1 plants-09-01011-f001:**
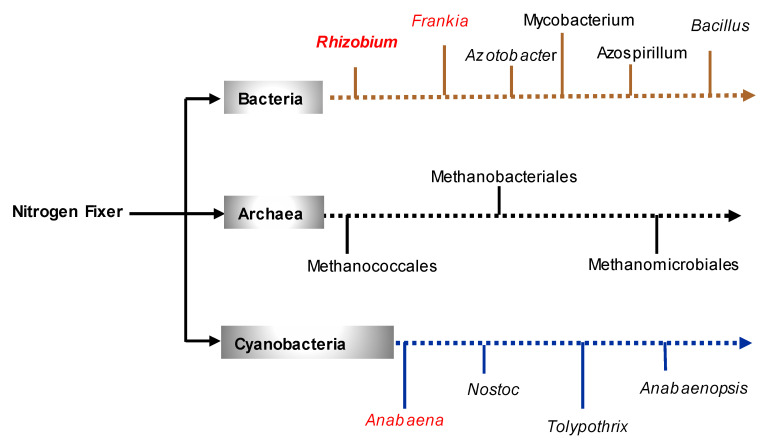
The three groups of nitrogen-fixing organisms including some main genera. In red: genera including symbiotic nitrogen-fixing species; in black: orders or genera (in italics) including free living nitrogen-fixing species.

**Figure 2 plants-09-01011-f002:**
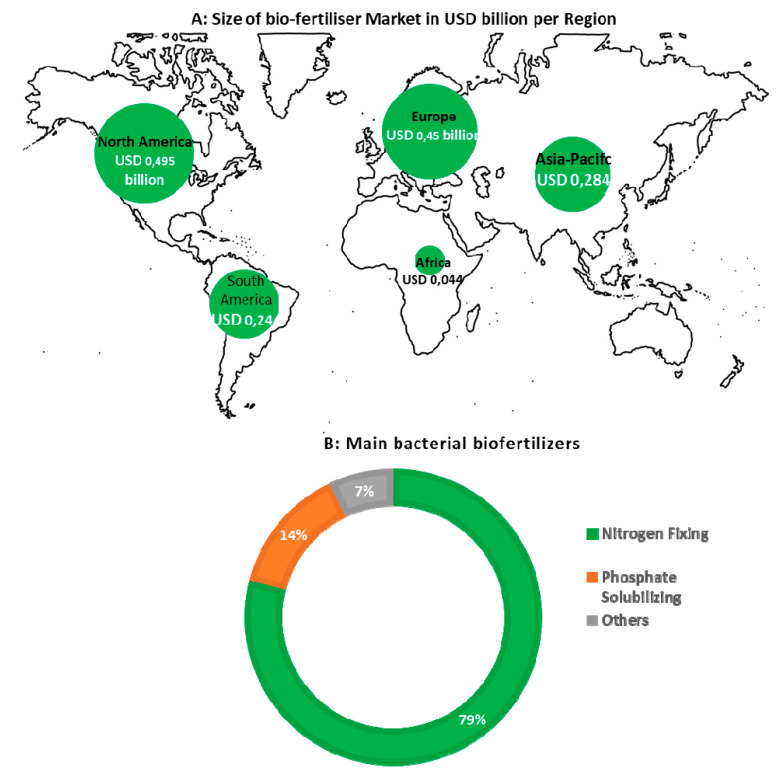
Global biofertilizers market and distribution. Data synthesized from Global Biofertilizers Market [84] and market data forecast [85]. (**A**): the diameter of each circle is proportional to the size of the market in USD billion. (**B**): percentage of main biofertilizers available in the market.

**Figure 3 plants-09-01011-f003:**
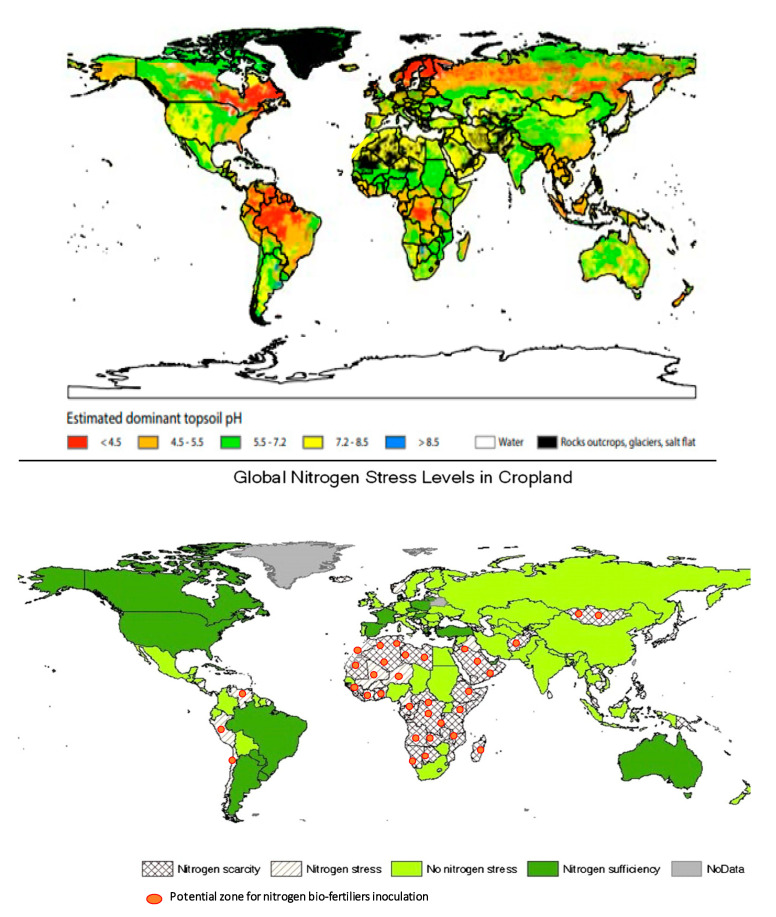
Potential zones for inoculation resulting from the superposition of nitrogen fertility and soil pH maps. Data synthesized from FAO/IIASA/ISRIC/ISS-CAS/jRC [170].

**Table 1 plants-09-01011-t001:** Some key dates of history of research on nitrogen, biological nitrogen fixation (BNF), and biofertilizers.

Date	Events	References
1836	Identification of nitrogen as a nutrient for plants	[75]
1886	Hellriegel and Wilfarth demonstrated the ability of legumes to convert N_2_	[20]
1888	First rhizobia were isolated from nodules	[64]
1893	Isolation of *Clostridium pasteurianum* (Free-living *N* fixers)	
1895	First commercial inoculant (Nitragen)	
1901	Isolation *Azotobacter*	[75]
1913	Carl Bosch performed Haber’s ammonia synthesis on an industrial scale	
1946	Second commercial inoculant (Azotogen)	[76]
1953	Identification of two nitrogen fixing bacteria: *Beijerinckia fluminensis* and *Azotobacter paspali*	[67,68]
1969	Positive results for ^15^N_2_ uptake by cyanobacteria	[71]
1972	Isolation of *Enterobacter cloacae* from corn roots	[77]
1975	Isolation of *Spirillum* sp. and demonstration of their nitrogenase activity	[78]
1984	Nitrogen fixation in Methanogens (archaea)	[74]
2011	The European Nitrogen Assessment provides the first integrated and comprehensive look at *N* use in Europe	[79]
2012	Database of all *nifH* sequences available in the Genbank nucleotide database	[80]

**Table 2 plants-09-01011-t002:** Some famous marketed microbe-based biofertilizers and target crops.

Name of Manufactured Products and Producer (in Italic)	Strain	Formulation	Crops Suited	Benefits According to the Authors	References
BioGro *Nguyen Thanh Hien in Hanoi University (Vietnam)*	*Pseudomonas fluorescens* *Bacillus subtilis* *Bacillus amyloliquefaciens* *Candida tropicalis*	Solid in peat	Rice (*Oryza sativa*)	Improve rice yield	[118]
Biofix *MEA company limited* *(Kenya)*	R*hizobium*	solid	-Soya bean (*Glycine max*) -Common bean (*Phaseolus vulgaris L*) -Alfalfa (*Medicago sativa)*	Cheaper than chemical nitrogen Lighter to transport, requires less labor effective for many crops	[109]
Bio N *Nutri-Tech Solutions* (*Australia*)	*Azotobacter* spp.	liquid	Horticulture	Access free atmospheric nitrogen. Increase yield and quality. Reduce soil erosion. Improve water retention. Enhance germination. Promote root growth. Phosphate release	[114]
Microbin and Azottein *Egyptian Ministry of Agriculture*	*Klebsiella*, *Bacillus*, *Azotobacter* *Azospirillum*	Carrier material	Barley cultivar Giza	Increased the different plant characteristic increases in grain yield reached approximately 24.8 and 27.2%	[92,104]
Legumefix *Legume Technology* *(UK)*	*Bradyrhizobium japonicum*	Sterile peat inoculant	Soybean and cowpea	grain yield (12–19%) relative to the control	[96]
Leguspirflo *SoyGro* *(South Africa)*	*Azospirillum brasilense*	Liquid	soybean	Inefficient	[97]
TerraMax’s Micro AZ product *TerraMax (Minnesota, USA)*	*Azospirillum brasilense* *A. lipoferum.*	Liquid	Wheat, Corn and Grain Sorghum	Improve root structure and stimulate root growth Provide biological nitrogen fixation Increases yields Stimulates rooting Increases yields	[119,120]
Nitrofix P *Agro-Input Suppliers Limited (AISL) (Malawi)*	*Bradyrhizobium japonicum* and *Bradyrhizobium elkanii*	Dry- inoculum based on gamma-sterilized peat	Soybeans	Promotes an increase in the yield by an average of 14.3–20.3% Reduced the nitrogen requirement	[79]
Vault LVL *BASF (Badische Anilin- & Soda-Fabrik) Germany*	*B. japonicum* + *Bacillus subtilis*	Liquid	Soybeans	Biomass yield improved	[98,100]

**Table 3 plants-09-01011-t003:** Summary of some successful steps achieved in nitrogen transfer from legumes to non-legumes.

Current Progress in Nitrogenase Transfer	References
*Nif* gene cluster transfer from nitrogen-fixing bacteria *Klebsiella pneumoniae* into *Escherichia coli*	[150]
Refactoring the nitrogen fixation gene to a simple cluster easy to access, engineering, and transferability.	[151]
*Nif* cluster gene transfer from nitrogen-fixing bacteria *Paenibacillus* sp. into *Escherichia coli*	[152]
Conception of an artificial FeFe nitrogenase system in *Escherichia coli*	[153]
Tranfer and expression of *Pseudomonas stutzeri* A1501 Nitrogen Fixation Island in *Escherichia coli*	[154]
Tranfer nitrogenase components (*nif*H) in *Saccharomyces cerevisiae* mitochondria (as model of eukaryotic cell)	[155]
Transgenic production of nitrogenase and expressing nitrogenase genes in plant plastids (tobacco as model)	[149]

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
