# Peer review of "Exploiting Biological Nitrogen Fixation: A Route Towards a Sustainable Agriculture"

_plants, 2020, doi:10.3390/plants9081011_

Round 1

Reviewer 1 Report

Presented manuscript is pointing on the history of nitrogen-fixing bacteria discovering, launching of the biofertilizers and their contribution for the modern agriculture. The review discusses some of the main problems for transferring of BNF to non legume plants and possible application of biofertilizers in the near future.

Although well written the manuscript suffers from the scientific point of view. The review combine large amount of well known facts with a few examples of BNF application in agriculture. But in the last case all examples, with small exceptions, are from different field experiments in Africa, never the less that on Fig 4 authors demonstrate that the bio-fertilizer market of Africa is only 3% from the global world market. Probably authors should correct the title of the review or to present more details also from Europe and America.

The scientific parts (2 and 5) are very general. A lot of publications from the last 2-3 years are missing, instead a number of other review articles are cited. In addition, part 5 was a topic of very recent review article in “Plants” (ref. 120).

The manuscript contains 5 figures and 3 tables, definitely too many for this kind of review. Figures are with low quality, some of them as Fig.1 and 2 should be omitted as they did not add anything new to the well known facts. Also Fig 5 (Global nitrogen stress levels) is confusing as it seems that there are no variations of the nitrogen stress levels inside the territory of the different country lands of Africa.  

Unfortunately also the tables are not precise. Table 3 for example:

a) bad formatting (in my copy there is even one empty row after Kefrifix.

b) missing information about the producers of biofertilizers. Definitely some of them are trade marks (Legumefix® for example).

c) authors should re-check all the plant names. Have to avoid using “Lucerne” while in the article they are talking about alfalfa. The Latin names (Medicago sativa in this case) of all crops are missing.

d) there are a lot of confusions with the references. For example according to the authors BioGro improve rice yield but the cited reference (83) is for soybean… Same for Legumefix and ref 84.

To my opinion also references are not adequate for this type of review. Lots of them are difficult to find and some are even not in English! But the main problem is that in a lot of cases they are not cited appropriately. For example:

Row 37 ref 2 is not dealing with world’s population but with the process of denitrification;

Row 71 ref 34 is a protocol;

Row 237 ref 72 and 73 should be 74 and 75 respectively;

Row 370 “Ivleva et al (122) but then in the references 122 is not Ivleva et al;

Row 372 Buren et all is missing the number etc. Authors should check very carefully all the references in the review and to use the appropriate ones!

Finally authors should choose better key words, what “nitrogen transfer” means?

There are also some other small corrections:

Row 127: “the process requires 28 ATP”, then in fig 1 ATP molecules are 12. But it is known that the correct number of ATP molecules in this process is 16!

Row 189. History of Nitrogen?

Row 215. What means: “North America (including US, Canada, Mexico and rest of North America)”? Further, US should be USA.

Row 218 Europe (including Germany, UK, Spain, Italy and France)?

Row 291. Bad citation. What 108 rhizobia / g of inoculant means?

Row 335. Authors are talking about “three different strategies” but latter only two strategies are presented.

Row 739 Ref 116 contains also ref 117 inside. Please correct!

Row 757 Oldroyd is G.E. not G.E.D.

My conclusion is that the presented review is an interesting work that after the main revision can be of interest for a vast audience.

Author Response

Dear Reviewers,

Find the answers to your questions in the attached file.

Reviewer 2 Report

I commend the authors for embarking on a review of this field however I am afraid to say that this review needs significant rewriting and refocusing to bring it up to what I think is a publishable standard. Exploiting BNF is a very large topic to tackle and one which it is not possible to do justice too in a minireview format.  The authors present an introduction into the field, before a somewhat perfunctory section on plant available nitrogen, followed by a brief mention of the background to NF, history of isolation of N2-fixing organisms, the briefest of mentions of the diversity of diazotrophs, the size of inoculant markets around the world (with the notable omittance of Australia and New Zealand) before a long section on the contribution of BNF to agriculture. The section on extending BNF to non-legumes seems out of place in this review as there is a very real dearth of detail provided. Finally, the authors end with a list research priorities to enable large-scale adoption of Nfixing inoculants, however many of these questions are not addressed in the review itself. In short, the review is poorly targeted, lacking depth and appropriate critique of the literature covered.

Specific points

  • The statement on line 56 is not possible to be supported by the literature.
  • Inconsistent use of Nitrogen, N-fixing, N2 (not N2)-fixing throughout manuscript. Not sure what authors mean in lines 81 and 82 relating to nitrate and ammonium.
  • Superscript and subscript missing on many chemical species. Incorrect labelling of nitrate and ammonium in figure 1.
  • Line 94 - unsupported value judgement
  • Figure 2: the accepted balanced equation for Nfixation is N2 + 8e- + 8H+ + 16ATP → 2NH3 + H2 + 16ADP + 16Pi
  • Statement on line 173 is misleading and needs to be reworded, as NfB was developed in 1975 by Fabio Pedrosa, not in 1901.
  • Paragraph starting line 247 to line 278 reads like a list of studies which are not appropriately critiqued
  • Line 279 - a very significant problem for adoption is the existence of high populations of resident soil rhizobia in soils targeted for inoculant delivery that outcompete the inoculant for nodulation of the host legume.
  • PGPRs appear briefly in line 305, but are these really the focus of this review?

Author Response

Dear reviewer 2,
Please, find the answers to your questions in the attached file.
We remain at your disposal for any other necessary information

Best regards

Reviewer 3 Report

The review article: ‘Exploiting Biological Nitrogen Fixation: a route towards a sustainable agriculture’ by Soumare et al. aimed to cover history of BNF research and highlight its agronomic importance as non-polluting way to improve crop production.

I commend the authors for their extensive data set divided into parts about nitrogen forms available for plants, comparison of BNF with chemical synthesis, organisms capable of N2 fixing and past-to-present trials to use them as biofertilizers, with emphasis on effective microbial products marketed to date. Authors comment on their efficiency and issues limiting their success in agriculture (problems with the formulation, strain viability, purity etc.). Authors emphasize that the successful adoption of BNF depends largely on understanding the field factors that control the BNF systems and improvement in the quality of commercial inoculants. In this context, the chapter ‘Road map for successful and large-scale adoption of N-fixing biofertilizers’ is not only well suited, but could also be little more detailed. The need for transferring the capacity to fix nitrogen to non-legumes/cereals is also highlighted as an attractive option (with challenges and possible future directions of cooperation mentioned) and opportunity to help underdeveloped countries.

The manuscript is clearly written in professional, unambiguous language. If there is any weakness, it is more editorial corrections to be made in the second half of the manuscript, which should be improved upon before acceptance. Questions for the future are well defined, relevant and meaningful. It is stated how the future research might fill the identified knowledge gap. Sufficient detail and specificity. The English language need not be improved substantially (only minor correction).

Author Response

Dear reviewer 3,
Please, find the answers to your questions in the attached file below.
We remain available for any other necessary information

Best regards

Reviewer 4 Report

This review covers a very wide scope and as a consequence is a relatively superficial overview of several different aspects of plant-growth-promoting inoculants and how they are thought to act. This inevitably means that there are many papers that could have been cited that were not, but also, in my view, some rather weak work was cited without critical assessment. There were also several points made that were either factually wrong or were misleading. I have highlighted some of these below along with some other comments.

  1. Lines 26 and 27. A good understanding of BNF ‘may’ allow its transfer to non-leguminous plants of high added value, but by stating this ‘will’ occur goes beyond what several experts believe possible.
  2. Line 56 states that N-fixing organisms are close to plant roots and this is true of both associative free-living N-fixers. The next sentence stating that the root nodule symbiosis is one of the most established flies in the face of the fact that associative free-living N-fixation is more widely established than the nodulation symbioses.
  3. Lines 65-67: the references here specifically relate to recent inoculant production for Africa without mentioning the background to inoculant production. Highly effective rhizobial and bradyrhizobial inoculants have been made in the USA, Europe and Australia starting about a hundred years ago as pointed out later in this review. Perhaps it might make more logical sense if it was made clear dearly on and in the abstract that this review is specifically looking at the developments and potential for Africa.
  4. Line 79 is factually inaccurate. Ammonia is NH3 and can be either soluble in water or gaseous. Ammonium NH4+ is protonated and since this proton comes from water it must only be in the soluble form, not the gaseous form as stated.
  5. Line 80 is misleading by omission and states that the organic forms of N are amino acids and proteins. This completely ignores the N present in bacterial polysaccharides like LPS and fungal polysaccharides like chitin, both of which are made up of polymers containing N-acetyl glucosamine.
  6. Line 86 cites Fig. 1 which incorrectly shows the transfer of soil organic matter to Organic nitrogen as being mineralisation. It is not, only the conversion from organic to inorganic forms is mineralisation.
  7. At the end of the first section (up to line107, the authors correctly point out the problems of chemical fertilizers and benefits of BNF and legumes. However they do not point out one of the problems of most legumes, which is that their net yield is much lower (usually less than half) of that of cereals. In terms of feeding the predicted population of 9 billion people, that is a very major problem. It has been estimated that if we were to rely only on BNF as a N input, the maximum world population would have to be limited to around 4 billion people (or less depending on different estimates). So the argument is much more nuanced than is put forward. There is a case that in countries with relatively poorly developed travel infrastructure, relying on BNF makes more sense. Why did the authors not look at Africa as a case in point?
  8. Line 121. The text states that BNF combines N2 with the hydrogen element. That is incorrect. As correctly indicated in Fig. 2 it uses hydrogen ions, or protons that come from water. Note in Fig. 2 that it states that 12 ATP are used to make 2 ammonia molecules, whereas line 27 states it takes 28 ATP. It is generally accepted that the minimal requirement is 16 ATP per N2 Furthermore, a necessary by-product of the reaction is H2, something omitted from the reaction shown. In this figure. water is shown as a product; it is NOT. The accepted reaction is usually written as:

N2 + 8 H+ + 8 e + 16 MgATP → 2 NH3 + H2 + 16 MgADP + 16 Pi

This is easily found, e.g. in Wikipedia and it is very worrying that the authors got this wrong. Furthermore, I could not understand where they got the value 675 kJ shown in the figure.

  1. It is stated (lines 173-4) that: ‘In 1901, the isolation of the aerobic heterotroph Azotobacter by Beijerinck was made possible by the development of a nitrogen free medium called NFb (Fb stands for Fabio Pedrosa)’. Pedrosa developed that medium in 1975 about 75 years after Beijerinck isolated Azotobacter.
  2. It is stated (Line 181 and in Table 1): Discovery of nitrogen fixation in cyanobacteria (formerly blue-green algae) has been established by Stewart in 1969 [67]. Nitrogen fixation in cyanobacteria was demonstrated in the 1930s, before Stewart was born.
  3. Figure 3: As pointed out by the authors on line 182, cyanobacteria were formerly called blue-green algae, a name that was dropped because, algae are eukaryotes and cyanobacteria are prokaryotes. Why then did the authors revert to the outmoded name blue-green algae in Fig. 3? Also in this figure, the genus names are not in italics, Nostoc is incorrectly spelled, and the legend refers to genera, but the archae names are actually orders not genera.
  4. Line 243 is incorrect stating: ‘N input through BNF is approximately 122 tons of N per year’; it should read million tons.
  5. Line: ‘sympal’ is mentioned with no explanation.
  6. Table 2 makes the point that with different inoculants soybean bradyrhizobial products stimulate grain yield by up to about 20%; that is in a well established legume system where growth stimulation is well documented. However Microbin and Azottein, with free-living fixers apparently in this study stimulate grain yield by around 25%. That sounds too good to be true, but the authors do not critically evaluate this. Critical assessment of yield tests would have been a valuable contribution, but in my view it is nor dealt with thoroughly in this review. Many of the studies cited refer to the potential for production of traits that could promote plant growth, but very few of these have been backed up with solid data on these traits from experiments with plant growth, either under greenhouse conditions or, more importantly, with field work.
  7. The paragraph in lines 349-354 states that since nitrogen fixation (nif) genes can be transferred to (and function in) coli there is now compelling proof that these objectives (transfer of N-fixation to non-legumes including cereals) can be achieved. This indicates to me that the authors do not really understand the challenges involved. It might just be possible, but many in the field remain sceptical that there will be any significant agricultural benefit.

Minor points

Throughout the manuscript the authors fail to use subscripts and superscripts properly e.g. as in N2 NH4+. They also fail to use italics for genus and species names as in Azotobacter, Rhizobium etc. and gene names like nifD and nifK. I am not sure if these are anomalies related to how text is submitted, but it seems unlikely because e.g. on lines 47 and 48 there are subscripts (one of which should be a superscript. Some terms such as bacteria and archae are incorrectly used as proper names starting with a capital letter. All this makes the manuscript appear very unprofessional.

Line 37: The population is predicted to reach 9 billion by 2050; the authors cannot know that it will.

Author Response

(The authors gave the same response as above.)

Reviewer 5 Report

Good and interesting review with an exhaustive survey of the biofertilizers containing nitrogen symbiotic and non-symbiotic bacterial inocula currently available on the market.

A few points need to be considered:

  • Line 17 in the abstract, I am not sure that unfortunately is the most appropriate term.
  • Line 26: the meaning of “share of nitrogen fixation in worldwide it not totally clear to me.
  • Line 48: what do you mean exactly by “mainly”.
  • Line 79: unless I missed something in the sentence, NH3 is a gas and not NH4+ which is soluble ammonium. Urea is an important source of fertilizer worldwide and it is not mentioned. It is however mentioned later line 92 and in Figure 2.
  • The section devoted to “Plant available nitrogen” is not really new as Figure 1 which can be found in large number of review and text books. Perhaps this section should be reduced to a minimum.
  • The same comment stands for the “Nitrogen fixation” section. Moreover, in this section, I am not sure that in Figure 2 the comparison of fertilizer production and nitrogen fixation in terms of energy cost is clearly understandable with respect to what is described in the text.
  • The description of the different biofertilizers is interesting, however in some places like line 217, it would have been better to specific if it is symbiotic or non-symbiotic nitrogen fixing bacteria or a mix of the two (see details listed in Table 2).
  • In the section “Challenge of extending the BNF ability to non-legumes”, the main findings at the physiological and molecular levels concerning the two possible approaches to engineer nitrogen fixation in non-legumes could have been developed a bit more. It will be thus more relevant when compared to the sections devoted to nitrogen assimilation which were quite extensively developed at the beginning of the review.
  • Perhaps, line 412 the recent review by Dellagi et al. 2020 J Exp Bot is worth to be cited.

  • Throughout the text put 15 in 15N in superscript.

Author Response

Dear reviewer 5,
Please find the answers to your questions in an attached file.
We remain available for any other necessary information

Best Regards

Round 2

Reviewer 1 Report

The revised version of the manuscript is much better. Authors replied in appropriate way to all my questions and remarks.

Author Response

Your comments and suggestions have significantly improved the review. The new version corrected according to the suggestions of other reviwers is attached to this answer

Reviewer 4 Report

Lines 368-370 the authors state: 'However, in view of the genetic advances and the successful transfer of nitrogen fixation (nif) genes to E. coli, there is now compelling proof that these objectives (transfer of N-fixation to cereals) can be achieved.

I do not agree that N-fixation between bacteria is 'compelling proof' that this can be transferred to plants.

Author Response

Lines 368-370 the authors state: 'However, in view of the genetic advances and the successful transfer of nitrogen fixation (nif) genes to E. coli, there is now compelling proof that these objectives (transfer of N-fixation to cereals) can be achieved.

I do not agree that N-fixation between bacteria is 'compelling proof' that this can be transferred to plants

We brought clarifications in the idea that we developed (Lines 379-380)